# A Knowledge Acquisition Method of Ship Coating Defects Based on IHQGA-RS

**Henan Bu \*, Xingyu Ji, Jiatao Zhang, Hongyu Lyu, Xin Yuan, Bo Pang and Honggen Zhou**

School of Mechanical Engineering, Jiangsu University of Science and Technology, Zhenjiang 212100, China; 199020051@stu.just.edu.cn (X.J.); rudolph163@163.com (J.Z.); just_202020025@163.com (H.L.); 192020002@stu.just.edu.cn (X.Y.); p2462079929@163.com (B.P.); yx18852893833@163.com (H.Z.)
\* Correspondence: buhn_just_edu@163.com

**Abstract:** Coating defects are caused by a series of factors such as the improper operation of workers and the quality of the coating itself. At present, the coating process of all shipyards is inspected and recorded at a specific time after construction, which cannot prevent and control defects scientifically. As a result, coating quality decreases, and production costs increase. Therefore, this paper proposes a knowledge acquisition method based on a rough set (RS) optimized by an improved hybrid quantum genetic algorithm (IHQGA) to guide the ship-coating construction process. Firstly, the probability amplitude is determined according to the individual position of the population, and the adaptive value k is proposed to determine the rotation angle of the quantum gate. On this basis, the simulated annealing algorithm is combined to enhance the local search ability of the algorithm. Finally, the algorithm is applied to rough set attribute reduction to improve the efficiency and accuracy of rough set attribute reduction. The data of 600 painted examples of 210-KBC bulk carriers from a shipyard between 2015 and 2020 are randomly selected to test the knowledge acquisition method proposed in the paper and other knowledge acquisition methods. The results show that the IHQGA attribute approximate reduction algorithm proposed in this paper is the first to reach the optimal adaptation degree of 0.847, the average adaptation degree is better than other algorithms, and the average consumption time is about 10% less than different algorithms, so the IHQGA has more vital and more efficient seeking ability. The knowledge acquisition result based on the IHQGA optimization rough set has 20–50% fewer rules and 5–10% higher accuracy than other methods, and the industry experts have high recognition. The knowledge acquisition method of this paper is validated on a hull segment. The obtained results are consistent with the expert diagnosis results, indicating that the method proposed in this paper has certain practicability.

**Keywords:** ships; painting defects; knowledge acquisition; rough sets; attribute reduction

## 1. Introduction

Various hull parts need to adopt different anti-corrosion measures because they are in different corrosive environments [1]. The correct coating process is the basis for ensuring its anti-corrosion performance. Marine coating includes the whole process of coating and surface treatment before coating, and coating uses suitable marine coatings to coat the ship's surface with the right technology to form a protective coating to prevent corrosion [2]. According to statistics, painting man-hours account for about 10%–20% of the ship construction man-hours, and the cost accounts for about 3%–8% of the total construction cost [3]. It can be seen that ship painting is an integral part of the shipbuilding process, and its quality is attracting more and more attention from shipyards and ship owners. At present, for large plate-type segments such as outer plates and flat bottoms, shipyards mainly use high-pressure airless sprayers for spraying operations. Various defects are produced during the spraying process due to workers' improper operation, environmental changes, and other factors, as shown in Figure 1a. Coating defects will affect the coating

performance and aesthetic appearance and cause severe corrosion on the hull surface. When the painting operation is completed, sanding the original paint for repainting in response to painting defects is also necessary, as shown in Figure 1b. This type of operation consumes many man-hours and increases the cost of shipbuilding, affecting the overall progress of shipbuilding and cost control.

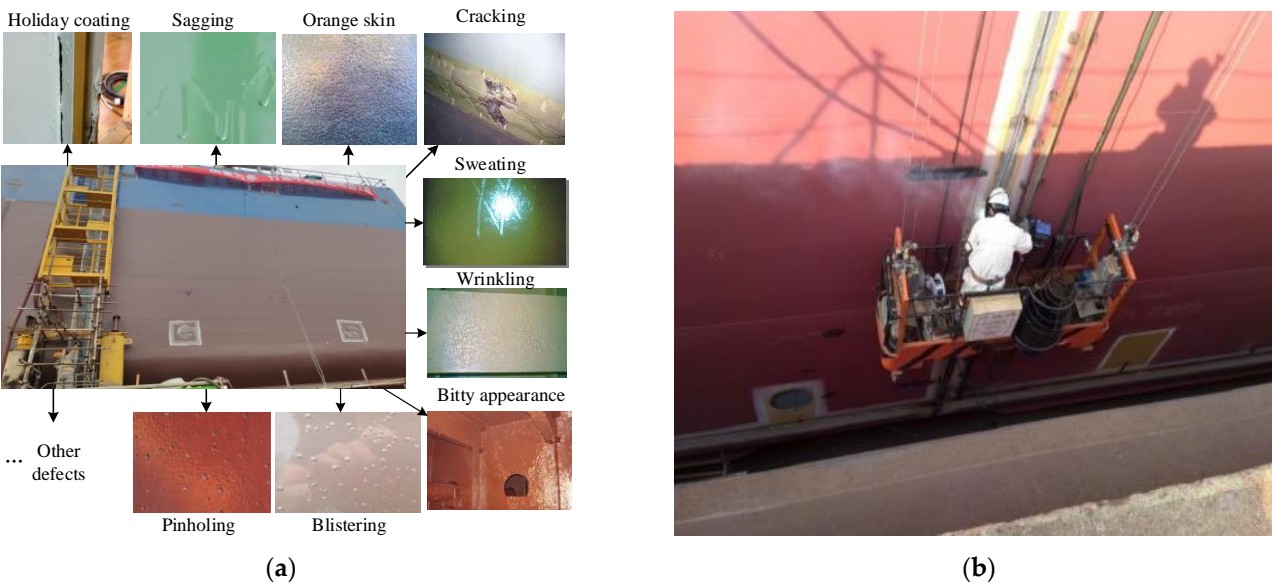

(**a**)  (**b**)

**Figure 1.** Ship painting operation site: (**a**) Painting defects site; (**b**) Painting and repairing site.

Nowadays, many intelligent robots and automated solutions for ship coating maintenance are available. Fernández-Isla et al. [4] proposed a new online vision inspection technique that can assist robots online in sandblasting in areas of coating damage. Muthugala et al. [5] proposed a novel energy-efficient Complete Coverage Path Planning (CCPP) method based on the Glasius Bioinspired Neural Network (GBNN) for a ship hull inspection robot. Li et al. [6] proposed an inexpensive semi-automated grit-blasting system to detect the rusted area and implement adaptive path planning for higher blasting efficiency. Prabakaran et al. [7] proposed that a robot is capable of navigating autonomously on the hull surface and performing water jet blasting to strip off the paint coating. The robot was also enabled with a Deep Convolutional Neural Network (DCNN)-based self-evaluating scheme that benchmarks cleaning efficiency. Abdulkader et al. [8] proposed an autonomous hull inspection robot that can navigate autonomously on the vertical metal surface, and it could perform metal thickness inspection.

The ship painting process involves a large amount of data information, including static data such as product models, process specifications, and process equipment information to support process design and planning and dynamic data acquired during the process implementation [9]. How to effectively use these data to provide guidance for coating designers and constructors and reduce the generation of coating defects is an inevitable requirement to improve coating quality and save coating costs. Knowledge engineering is the process of accumulating, sorting out, passing on, sharing, and reusing knowledge in the field to make it available to everyone and improve the quality of people's work [10]. Knowledge acquisition is the core step in knowledge engineering, which can acquire potential knowledge from an extensive, incomplete, and ambiguous dataset, including interactive knowledge acquisition and automatic knowledge acquisition forms [11]. Interactive knowledge acquisition means summarizing and refining knowledge through human or human–computer communication. In contrast, automatic knowledge acquisition means automatically refining new knowledge from datasets that have not yet been formalized or even discovered through a computer program with advanced learning functions [12].

Research results have been conducted in China and abroad for automatic knowledge acquisition. Akgobek et al. [13] proposed a rule extraction algorithm (REX-1) algorithm for automatic knowledge acquisition in inductive learning. Montanari et al. [14] present an algorithm for obtaining health information from electrical asset components. Zhang et al. [15] used an adaptive convolutional neural network to extract building information from satellite images. Rani et al. [16] proposed a tumor-sensing algorithm to diagnose cancer cells by preprocessing, segmentation, and alternative methods. Zhang et al. [17] propose a parallel method for computing rough sets' upper and lower approximations, combined with MapReduce technology for massive data mining. Ye et al. [18] proposed a novel multi-level coarse set model (MLRS) based on attribute value taxonomies (AVT) and a complete subtree promotion scheme for mining data with attribute value classification. Qu et al. [19] mined the investment decision knowledge of water projects through rough set theory for investment risk assessment. Agarwal et al. [20] used rough sets for the mining grinding process to investigate the effect of its various input parameters on the response. All the above studies have achieved good application results and provided an excellent theoretical basis for acquiring ship painting defects.

Rough set theory is a mathematical tool for dealing with uncertainty proposed by the Polish scientist Z. Pawlak in 1982 [21]. The main idea is to divide the data and deal with uncertain or incomplete information and knowledge quantitatively and analytically. Rough sets have received much attention as an effective intelligent information processing technique [22]. Attribute simplification is the core of the rough set theory, and its role is to eliminate redundant attributes or redundant features and play the role of dimensionality reduction. Many scholars have studied the attribute simplification algorithm. Wei et al. [23] proposed an attribute approximation algorithm based on fast extraction and multi-strategy social spider optimization. Ding et al. [24] proposed a new multi-granularity super-trust fuzzy rough set-based attribute approximation (MSFAR) algorithm to address the problem of a large amount of uncertainty regarding unstructured and imprecise data in extensive data analysis. Xie et al. [25] proposed a heuristic attribute approximation algorithm based on the binary bat algorithm. Zhang et al. [26] proposed a hybrid approach based on generalized gray correlation analysis (GGRA) and decision experiment and evaluation laboratory (DEMATEL) for attribute approximation. Liu [27] used a discriminable matrix-based approach to study the attribute approximation problem. The above attribute reduction methods enhance the performance to some extent, but there are still problems such as low reduction efficiency and low and poor reduction accuracy.

Therefore, this paper proposes an improved hybrid quantum genetic algorithm (IHQGA) to integrate the simulated annealing algorithm (SA) with the quantum genetic algorithm. The IHQGA improves the local search ability on the powerful global search ability of the quantum genetic algorithm and uses it in the rough set attribute reduction process to improve the efficiency and accuracy of attribute reduction approximation. Knowledge acquisition is carried out for a shipyard painting defect example, and the effectiveness and feasibility are verified by comparing it with other algorithms.

## 2. A Knowledge Acquisition Method Based on Rough Set Theory

### 2.1. Rough Sets Theory Preliminaries

This section will introduce the main parts of classical rough set theory, including its basic model and the concept of attribute reduction.

#### 2.1.1. Information Systems

In rough set theory, an information system is used to represent knowledge. The information system can be expressed as $S = (U, A)$, where $U$ is a set of nonempty, finite, and global individuals, and $A$ is a set of nonempty and limited attributes; that is, for attribute $a \in A$, there is $a : U \to V_a$, where $V_a$ is the set of values of $a$, which is called the range of $a$ [28].

### 2.1.2. Decision Systems

For information system $S = (U, A)$, let $C$ and $D$ be two subsets of the attribute set $A$ and call $C$ and $D$ conditional attributes and decision attributes of $A$, respectively. In this way, $S$ is represented as $T = (U, C \cup D)$, which is a decision system [29].

### 2.1.3. Attribute Reduction

For a decision system, there is often some degree of association or dependence between the attributes in $C$. Some features may be redundant compared with attribute set $D$. Attribute reduction is to replace the original conditional attribute set with the most straightforward condition attribute set and still judge the decision attributes according to the remaining condition attributes without losing any information of the decision system [30].

### 2.2. Knowledge Acquisition Steps Based on Rough Set

The rough set theory can directly analyze and reason with the data to discover the implied knowledge and reveal the potential laws, a natural knowledge discovery method. Therefore, this paper carries out the knowledge acquisition of ship painting defects based on rough set theory and proposes an improved hybrid quantum genetic algorithm to optimize its attribute reduction process, the flow of which is shown in Figure 2. The process of ship painting defect knowledge acquisition based on rough set theory is shown in Figure 2.

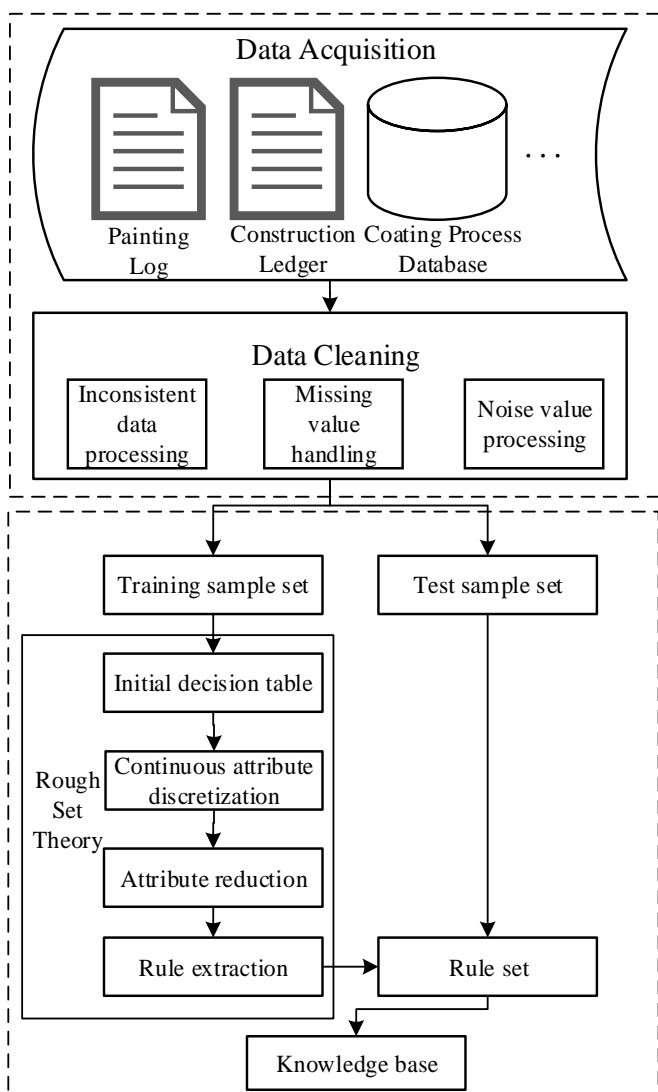

**Figure 2.** Knowledge acquisition process of ship painting defects based on rough set theory.

The specific steps are as follows.

Data Acquisition: The painting defect data are collected through painting logs, construction ledgers, and a painting process database in the shipyard. In this paper, for unstructured data such as various reports and pictures, the relevant data are manually sorted and filtered from sources such as painting logs and construction ledgers; for structured data such as temperature, wind speed, and paint information, they are retrieved from sources such as the painting process database, shipyard ERP (Enterprise Resource Planning) system, and sensors.

Data Cleaning: Data cleaning includes inconsistent data processing, missing value processing, and discrete value processing, which is an indispensable part of the whole data analysis process, and the quality of its results is directly related to the model effect and conclusion. In this paper, manual modification and computer program correction are applied for inconsistent data according to the specific situation. The most probable value is used for missing value handling to fill in the missing values for attributes with discrete values. The mean value is used to fill in the missing values for attributes with continuous values. For noisy data, the dataset is boxed by the number of rows of records using the equal depth box method and smoothed by the mean value using the data-smoothing method.

Sample training: The dataset obtained in the previous step is divided into a training sample set and a test sample set. The training sample set is used to train the model using the data from the training set, and then, the test set is used to verify the final effect of the rough set knowledge acquisition.

Build the initial decision table: According to the training sample set, the attributes are scored using the Pearson coefficient, and the attributes are selected according to their relevance for defects, reducing the complexity of rough set knowledge acquisition, dividing the decision attributes and conditional attributes, and constructing the initial decision table $S$.

Continuous property discretization: Since the information related to coating defects are mostly continuous values, the fuzzy c-mean clustering algorithm (FCM) is used to discretize process information, environmental information, and parameter information one by one, and finally, the decision table $S'$ is obtained.

Attribute reduction: In this paper, we propose an improved hybrid quantum genetic algorithm for rough set attribute reduction, improve the performance of the algorithm by improving the taking of probability amplitude and rotation angle coefficient k in quantum genetic algorithm, and finally fuse the simulated annealing algorithm to enhance the local search ability to avoid premature convergence.

Rule extraction: The rules are extracted from the minimal decision table after attribute reduction, the same rules are deleted, and the minimal rules are stored.

Knowledge base rule storage: The rule knowledge extracted based on the training set will be examined against the test set objects to verify the generated rules' validity and accuracy. Finally, the correct set of rules is saved to the knowledge base.

## 3. Attribute Reduction Based on IHQGA

The introduction states that attribute reduction is the core of the rough set theory. There are traditional attribute reduction algorithms such as the positive domain-based reduction method and the difference matrix-based reduction method. The traditional reduction algorithms lead to exponential time complexity as the number of objects increases [31]. With the emergence of various advanced algorithms, scholars have applied advanced algorithms to attribute approximation, such as the literature mentioned above [16,18], etc. Genetic algorithms have also been used in attribute approximation with some success, but there is still the problem of low efficiency [32]. Therefore, in this paper, we use a quantum genetic algorithm with a simulated annealing algorithm for attribute approximation and improve the algorithm to enhance its performance further.

*3.1. Principle of Quantum Genetic Algorithm*

The QGA (Quantum Genetic Algorithm) is a new algorithm based on GA (Genetic Algorithm) introducing quantum computing theory [33]. It uses quantum coding and performs a population update by the quantum gate update strategy. The essence of the algorithm lies in quantum evolution and the group search method, so it has powerful global search ability and high operational efficiency. Compared with the traditional GA, QGA is superior, but QGA still has shortcomings such as more iterations when optimizing complex functions and falling into local extremes [34,35].

3.1.1. Quantum Bit Encoding

Unlike the coding of the traditional GA, QGA uses quantum bits for coding, and chromosomes are represented by quantum bits. In general, the state of a quantum bit can be either 0 or 1, or a linear superposition of both, so its shape can be expressed as:

$$|\Psi\rangle = \alpha|0\rangle + \beta|1\rangle \tag{1}$$

where $\alpha$ and $\beta$ are the probability amplitudes of $|0\rangle$ and $|1\rangle$, respectively, and satisfy the normalization conditions:

$$|\alpha|^2 + |\beta|^2 = 1. \tag{2}$$

Thus, a quantum bit chromosome of length $m$ can be expressed in the following form:

$$q_j^t = \begin{bmatrix} \alpha_{j1}^t & \alpha_{j2}^t & \cdots & \alpha_{jm}^t \\ \beta_{j1}^t & \beta_{j2}^t & \cdots & \beta_{jm}^t \end{bmatrix} \tag{3}$$

where $q_j^t$ denotes the $j$-th individual of the $t$-th generation.

In Equation (3), $\alpha_{ji}^2 + \beta_{ji}^2 = 1$, $i = 1, 2, \cdots, m$. This coding method can represent the quantum superposition state; for example, there is a 3-bit quantum system, and its probability amplitude is as follows:

$$\begin{bmatrix} \frac{1}{\sqrt{2}} & \frac{1}{2} & \frac{1}{\sqrt{2}} \\ \frac{1}{\sqrt{2}} & \frac{\sqrt{3}}{2} & \frac{1}{\sqrt{2}} \end{bmatrix} \tag{4}$$

The system can be described by the following superposition:

$$\frac{1}{4}|000\rangle + \frac{1}{4}|001\rangle + \frac{\sqrt{3}}{4}|010\rangle + \frac{\sqrt{3}}{4}|011\rangle + \frac{1}{4}|100\rangle + \frac{1}{4}|101\rangle + \frac{\sqrt{3}}{4}|110\rangle + \frac{\sqrt{3}}{4}|111\rangle \tag{5}$$

Equation (5) indicates that the occurrence probabilities of states $|000\rangle$, $|001\rangle$, $|010\rangle$, $|011\rangle$, $|100\rangle$, $|101\rangle$, $|110\rangle$, and $|111\rangle$ are 1/16, 1/16, 3/16, 3/16, 1/16, 1/16, 3/16, and 3/16, respectively. Therefore, the system represented by Equation (4) can represent information of eight states at the same time.

Therefore, the evolutionary computation using qubit representation has better diversity than the traditional method. As shown in Equation (4), only one quantum chromosome can represent eight states, which requires at least eight chromosomes in the conventional representation.

3.1.2. Quantum Revolving Gate Update

The algorithm selects individuals with better fitness through population updating. This process is completed by using the quantum revolving gate. The quantum revolving gate is the primary operator of population updating. A standard quantum gate is:

$$U(\Delta\theta) = \begin{bmatrix} \cos\Delta\theta & -\sin\Delta\theta \\ \sin\Delta\theta & \cos\Delta\theta \end{bmatrix} \tag{6}$$

where $\theta$ is the rotation angle. The population renewal process can be expressed as:

$$\begin{bmatrix} \alpha'_i \\ \beta'_i \end{bmatrix} = \begin{pmatrix} \cos(\theta_i) & -\sin(\theta_i) \\ \sin(\theta_i) & \cos(\theta_i) \end{pmatrix} \begin{bmatrix} \alpha_i \\ \beta_i \end{bmatrix} \tag{7}$$

In a word, the key of quantum genetic algorithm lies in quantum encoding and decoding and the determination of quantum gate and its rotation angle.

### 3.2. Design of IHQGA

The quantum genetic algorithm and annealing algorithm are both probabilistic searches for optimal search. The quantum genetic algorithm encodes the chromosome probability and uses the iterative genetic factor, quantum gate update heuristically adaptive search to the optimal solution, but it is easy to fall into local extremes, which affects the operation speed. The simulated annealing algorithm, on the other hand, combines the probabilistic burst-hopping property to randomly search for the optimal solution in the solution space at a specific initial temperature, along with the decreasing temperature parameter, which has a solid local search capability and enables the search process to avoid falling into local optimal solutions. In this paper, according to the advantages and disadvantages of the two algorithms, the simulated annealing algorithm is introduced in the search process of the quantum genetic algorithm, and the probability amplitude and quantum rotation angle of the quantum genetic algorithm are improved to improve the performance of the algorithm further.

The steps of the IHQGA design for attribute reduction are as follows.

(1)　Determination of probability magnitude

In the traditional quantum genetic algorithm, in the process of initializing the population, the qubits of all chromosomes are equal probability superposition, that is, $\alpha^t_{ji}$, $\beta^t_{ji}$ (i = 1, 2, $\cdots$ , m), which takes $1/\sqrt{2}$. This balanced superposition initialization method is undoubtedly reasonable in search algorithms such as Grover, but there are many disadvantages in evolutionary algorithms. The individuals in the initial population are the same quantum coding, which slows down the convergence speed of quantum gate update. This paper takes different initial values for $\alpha$ and $\beta$ according to the individual position of the population:

$$\alpha_{ji} = \sqrt{\frac{j}{n}}, \; \beta_{ji} = \sqrt{\frac{n-j}{n}}, \; j = 1, 2, \cdots , n, \; i = 1, 2, \cdots , m \tag{8}$$

(2)　Determination of rotation angle of quantum gate

The QGA updates population chromosomes mainly through operations such as Equation (6) quantum rotatory gates. As the main parameter of quantum rotatory gates, $\theta$ is the key to the implementation of QGA. A pre-designed adjustment strategy generally determines the size and symbol of rotation angle $\theta$. The symbol determines the direction of convergence, and the magnitude affects the convergence rate.

In IQGA, the rotation angle can be expressed as $\theta = k \times f(\alpha, \beta)$, where $k$ is a factor related to the convergence rate of the algorithm. If the value of $k$ is too large, the search range of the algorithm is extensive, and the jump is too fast, it is easy to omit the better solution, resulting in the convergence of the algorithm to the local extreme point. Conversely, the jump in the search range of the algorithm is too small, the search speed is too slow, and the algorithm is prone to stagnation. So, in this paper, $k$ is defined as an adaptive variable related to evolutionary algebra so that $k$ can adjust the search range of the algorithm adaptively according to the number of iterations of the algorithm. Set $k = 10 \times \exp(-t/\max t)$, where $t$ is the iteration algebra and $\max t$ is the maximum evolutionary algebra. The function $f(\alpha, \beta)$ determines the convergence direction of the algorithm and ensures that the rotary door operation converges toward the optimal solution.

(3)    Determination of fitness function

The fitness function design is a critical link in all kinds of genetic algorithms, which affects the convergence speed and efficiency of the algorithm. The fitness function of the quantum genetic algorithm is designed according to the problem of attribute reduction in the rough set. Given the initial decision table $S = (U, C \cup D, V, f)$, $C$ is the conditional attribute set, $D$ is the decision attribute set, $V$ is the value domain of the attribute set, and $f$ is the information function. For individual $P_j$ in population $P$, the fitness function is constructed as follows:

$$f(P_j) = 1 - \frac{|p_j|}{|C|} \cdot \frac{1}{e^{\gamma_C(D) - \gamma_{P_j}(D)}} \tag{9}$$

where $|P_j|$ is the number of conditional attributes contained by the individual, $|C|$ is the total number of conditional attributes in the initial decision table, $\gamma_C(D)$ is the classification quality of conditional attributes to decision attributes in the decision table, as shown in Equation (10), and similar $\gamma_C(D)$ is the approximate classification quality of individual $P_j$ to decision attributes.

$$\gamma_C(D) = \frac{\left| \underset{X_i \in U|D}{\cup} C - (X_i) \right|}{|U|} \tag{10}$$

(4)    Enhanced local search based on SA

Due to the local search ability of simulated annealing algorithm, this paper leads it into simulated annealing algorithm in the population evolution process of quantum genetic algorithm. Suppose there is a convergent population and the search accuracy is not reached. In that case, a simulated annealing algorithm can help it jump out quickly to help the quantum genetic algorithm find the optimal population update direction and speed up the search speed. The principle is shown in Figure 3. In the figure, $Q(t-1)$ is the initial population initialized by the probability magnitude and $q_1^{t-1}$ is an individual in the initial population. The probability code of $Q(t-1)$ is converted into a binary string to solve for fitness and stored in $P(t)$, where $P(t) = \{x_1^t, x_2^t, \cdots, x_m^t\}$ and $x_j^t (j = 1, 2, \cdots, m)$ are binary strings. $P(t)$ is subjected to a simulated annealing algorithm to find the optimal population $\overline{P(t)}$, and the individuals in $\overline{P(t)}$ are brought into the fitness function to solve for the fitness value, and the optimal population $B(t)$ is set. The algorithm ends when $B(t)$ converges and meets the requirements of the problem; otherwise, the population is updated to continue the optimization search, and the evolution of the population $Q(t)$ to $Q(t+1)$ in the algorithm is updated using the quantum gate rotation strategy in Section 3.2 (2). Each population generation needs to go through the SA to find the best, and the individuals updated by the SA continue to perform the remaining steps of the QGA.

### 3.3. The Flowchart of IHQGA

In this paper, a hybrid simulated annealing-quantum genetic algorithm (IHQGA) is proposed for the rough set attribute reduction process. The flow of IHQGA is shown in Figure 4.

The main steps of this algorithm are as follows.

STEP1: Each entry in the decision table is quantum bit encoded.

STEP2: Initialize the algorithm parameters and obtain the initial population $Q(t)$ according to Equation (7).

STEP3: The population individuals are measured to convert their probability codes into binary number strings to solve for fitness and obtain the population $P(t)$.

STEP4: After measurement, each individual fitness is solved according to the fitness function of Equation (9), and $B(t)$ is set as the optimal population to deposit the optimal individuals.

STEP5: Judgment on the optimal population. If it converges, go to the next step for simulated annealing; if it does not converge, go to STEP11.

STEP6: The optimal individual in $B(t)$ is used as the initial solution $q$, and its corresponding objective function value is calculated.

STEP7: Apply a perturbation to the initial solution $q$ to generate a new solution $q'$ and calculate its corresponding objective function value.

STEP8: If the function value of the new solution is greater than or equal to the function value of the initial solution, then accept the new solution as the current solution; otherwise, accept the new solution according to the Metropolis criterion.

STEP9: Determine if the number of iterations is reached, and if it reaches the number of iterations, go to the next step; otherwise, go to STEP7.

STEP10: Determine if the termination condition is satisfied. If it is satisfied, the current optimal solution will be the next generation and go to the next step; otherwise, reset the number of iterations and go to STEP7.

STEP11: Update the population using the quantum gate update strategy in Section 3.1 (2) to obtain the population $Q(t + 1)$.

STEP12: Perform the measurement and fitness calculation in the same way as above to obtain the population $P(t + 1)$.

STEP13: Compare $B(t)$ and $P(t + 1)$ to deposit the optimal individuals into $B(t + 1)$.

STEP14: Determine whether it meets the search requirement or the maximum number of iterations. If yes, end; otherwise, return to STEP5.

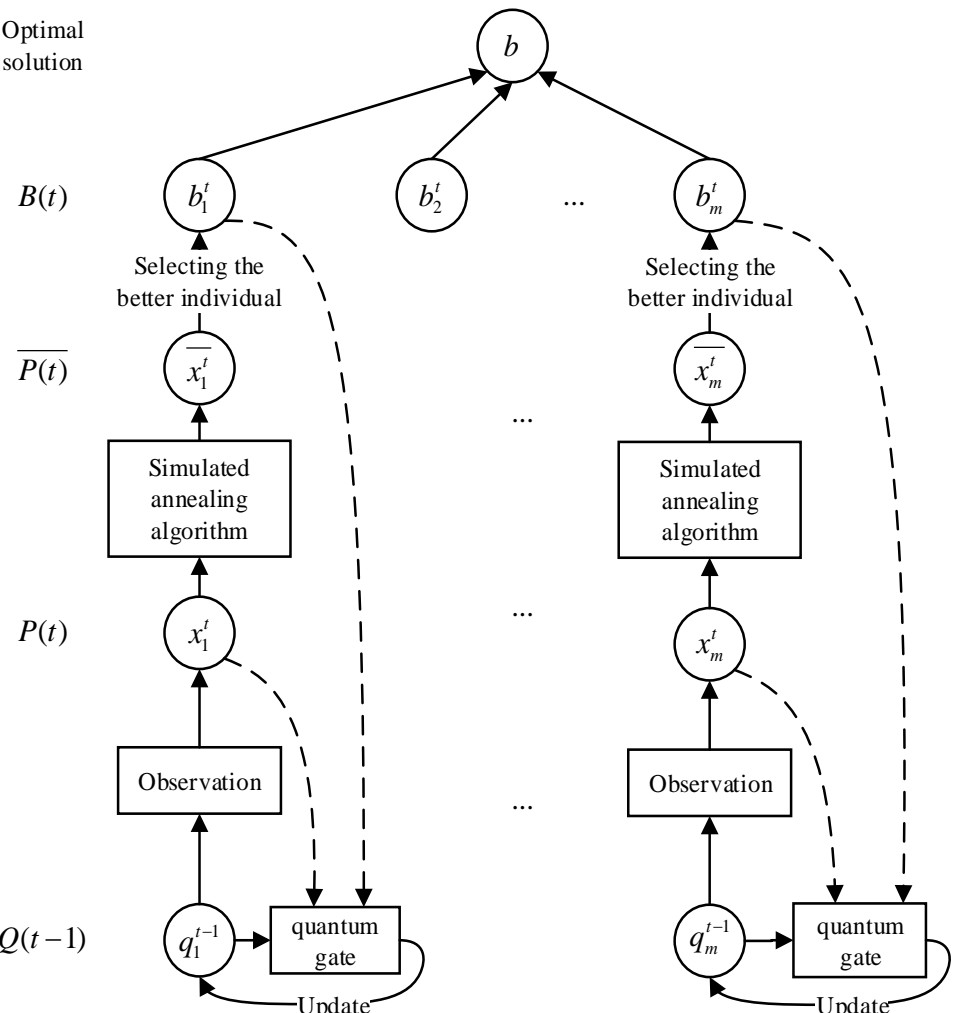

**Figure 3.** Principle of QGA combined with SA.

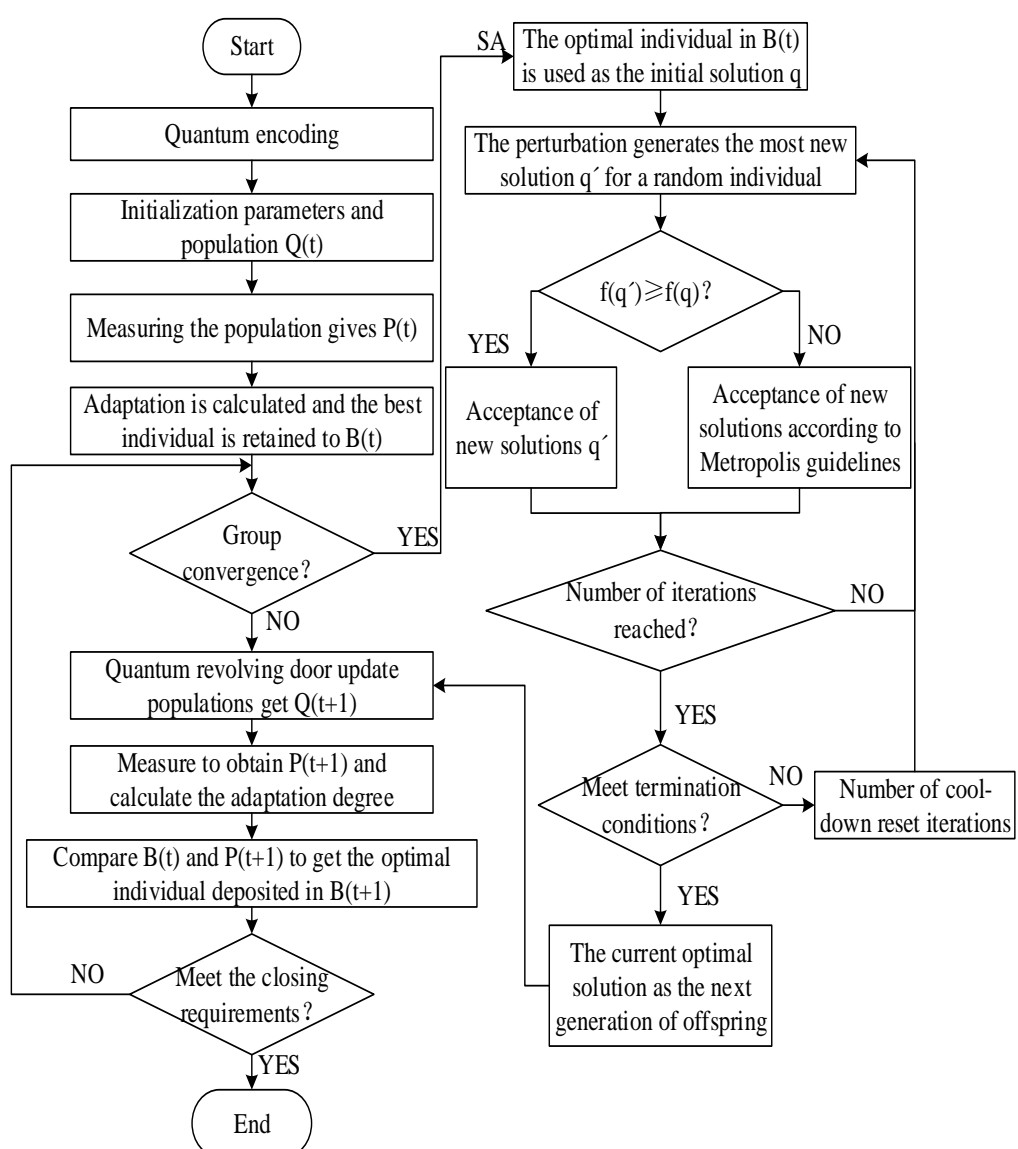

**Figure 4.** Flowchart of improved hybrid quantum genetic algorithm.

## 4. Results Analysis and Models Comparison

### 4.1. Data Preparation

Knowledge acquisition of marine painting defects helps to explore the law of defect formation to effectively guide the painting designers and constructors to avoid the generation of painting defects. The source of the experimental dataset is the painting examples of the 210-KBC bulk carrier painting process using an NJ6528 high-pressure airless spraying machine in a shipyard between 2015 and 2020. Each example contains painting process information, equipment parameter information, operating environment parameters, and defect records. The painting defect dataset is extracted through the painting process database and relevant construction documents. The dataset is pre-processed to obtain 600 initial data, which are shown in Table 1.

**Table 1.** Example data of ship coating defects.

| Examples | 1 | 2 | 3 | 4 | ... | 597 | 598 | 599 | 600 |
|---|---|---|---|---|---|---|---|---|---|
| Coating position | Cabin | Deck | Bottom | Bottom | ... | Deck | Deck | Bottom | Waterline |
| Spraying pressure (MPa) | 15 | 18 | 25 | 20 | ... | 20 | 22 | 22 | 25 |
| Spraying distance (cm) | 20 | 12 | 20 | 25 | ... | 15 | 18 | 24 | 25 |
| Coating viscosity (Pa·s) | 237 | 285 | 330 | 320 | ... | 130 | 120 | 200 | 125 |
| Wet film thickness (μm) | 44 | 65 | 125 | 58 | ... | 130 | 135 | 48 | 130 |
| Paint type (drying speed) | Slow drying | Standard drying | Fast curing | Fast curing | ... | Standard drying | Standard drying | Fast curing | Standard drying |
| Rusting grade | Sa2 | Sa2 | Sa2 | Sa2 | ... | Sa2 | Sa2 | Sa2 | Sa1 |
| Wind grade | 2 | 1 | 2 | 6 | ... | 5 | 5 | 5 | 2 |
| Atmospheric temperature (°C) | 18 | 24 | 16 | 29 | ... | 24 | 36 | 24 | 24 |
| Relative humidity (%) | 65 | 55 | 55 | 80 | ... | 35 | 55 | 25 | 35 |
| Surface temperature (°C) | 16 | 17 | 7 | 26 | ... | 10 | 24 | 15 | 10 |
| Work team | Group C | Group B | Group A | Group A | ... | Group A | Group A | Group B | Group A |
| Coating type | Chlorinated rubber paint | Pure epoxy paint | Anti-fouling paint | Anti-fouling paint | ... | Rust preventive paint | Rust preventive paint | Anti-fouling paint | Anti-corrosive paint |
| ... | ... | ... | ... | ... | ... | ... | ... | ... | ... |
| Defect name | Holiday | Holiday | Sagging | Sagging | ... | Blistering | Blistering | Dry spraying | Fish-eye |

The Pearson coefficient is used for the feature selection of the initial data table. Among them, discrete features were given their rank ranking according to the actual production situation on site, as shown in Table 2. The correlation coefficient of each attribute to the target value and its *p*-value (significance value) were calculated by Statistical Product and Service Solutions (SPSS) software, and the features with moderate and robust correlation and significance were retained, as shown in Table 3. The absolute values of correlation coefficients were not correlated between 0 and 0.1, weakly correlated between 0.1 and 0.3, moderately correlated between 0.3 and 0.5, and strongly correlated above 0.5; *p*-values less than 0.05 indicated significant correlation and more than 0.05 indicated insignificant correlation.

Finally, 12 features were obtained and used as conditional attributes, and 17 painting defects were used as decision attributes. For the convenience of representation, the conditional attributes and decision attributes were coded, and the coding results are shown in Table 4. The conditional attributes and decision attribute generated the initial decision table, and the continuous attributes were discretized using the fuzzy C-mean clustering algorithm (FCM). The categorical variables were mapped to form the values, and the decision table after discretization is shown in Table 5. The decision attribute values 1 to 17 in the table correspond to decision attribute d. The meanings of the conditional attribute values are shown in Table 6.

**Table 2.** Discrete feature ranking.

| Grade Name | 1 | 2 | 3 | 4 | 5 | 6 | 7 | 8 | 9 |
|---|---|---|---|---|---|---|---|---|---|
| Coating position | Cabin | Superstructure | Bottom | Outboard | Waterline | Deck | — | — | — |
| Coating type | Pure epoxy paint | Coal tar epoxy paint | Vinyl tar paint | Inorganic zinc silicate | Chlorinated rubber paint | Self-polishing co-polymer | Anti-fouling paint | Rust preventive paint | Anti-corrosive paint |
| Work team | Group A | Group B | Group C | — | — | — | — | — | — |
| Paint type (drying speed) | Fast curing | Slow drying | Standard drying | — | — | — | — | — | — |

**Table 3.** Correlation coefficients and significant values.

| Examples | Correlation Coefficient (Absolute Value) | *p*-Value (Significance Value) |
|---|---|---|
| Coating position | 0.526 | 0.032 |
| Spraying pressure | 0.825 | 0.015 |
| Spraying distance | 0.598 | 0.031 |
| Coating viscosity | 0.722 | 0.027 |
| Wet film thickness | 0.342 | 0.045 |
| Paint type (drying speed) | 0.389 | 0.045 |
| Rusting grade | 0.750 | 0.024 |
| Wind grade | 0.412 | 0.041 |
| Atmospheric temperature | 0.919 | 0.006 |
| Relative humidity | 0.847 | 0.012 |
| Surface temperature | 0.388 | 0.044 |
| Work team | 0.992 | 0.001 |
| Coating type | 0.057 | 0.271 |

**Table 4.** Condition attribute and decision attribute code.

| Type | Code | | | | | |
|---|---|---|---|---|---|---|
| | **a1** | **a2** | **a3** | **a4** | **a5** | **a6** |
| Condition properties | Coating position | Spraying distance | Spraying pressure | Coating viscosity | Wet film thickness | Paint type (drying speed) |
| | **a7** | **a8** | **a9** | **a10** | **a11** | **a12** |
| | Rusting grade | Wind grade | Air temperature | Relative humidity | Surface temperature | Work team |

| Type | Code | | | | | |
|---|---|---|---|---|---|---|
| | **d1** | **d2** | **d3** | **d4** | **d5** | **d6** |
| Decision attribute | Holiday | Sagging | Orange skin | Dry spray | Pinhole | Blistering |
| | **d7** | **d8** | **d9** | **d10** | **d11** | **d12** |
| | Fish eye | Wrinkling | Exudation | Blushing | Water bubbling | Corrosion |
| | **d13** | **d14** | **d15** | **d16** | **d17** | - |
| | Cracking | Peeling | Chalking | Bleeding | Pin holes | - |

*4.2. Experimental Results and Discussions*

The above 1/3 coating defect data are selected as the test set and 2/3 are selected as the training set, and the traditional quantum genetic algorithm-based attribute reduction algorithm [36], the genetic algorithm-based interval-valued attribute reduction algorithm (ARIGA) [37], and the IHQGA algorithm proposed in this paper are used for attribute reduction. The three algorithms of ARIGA, QGA, and IHQGA are analyzed for population diversity, merit-seeking ability, and reduction. Then, the knowledge acquisition results of this paper are compared with other methods to verify the effectiveness of knowledge acquisition. Finally, the practicability of this method is verified by a segment in a project example.

**Table 5.** Decision table after discretization.

| U | a1 | a2 | a3 | a4 | a5 | a6 | a7 | a8 | a9 | a10 | a11 | a12 | d |
|---|----|----|----|----|----|----|----|----|----|-----|-----|-----|---|
| 1 | 1 | 0 | 1 | 0 | 0 | 1 | 1 | 0 | 1 | 2 | 1 | 2 | 1 |
| 2 | 4 | 1 | 0 | 1 | 1 | 1 | 0 | 0 | 1 | 1 | 1 | 2 | 1 |
| 3 | 3 | 2 | 1 | 1 | 2 | 2 | 3 | 0 | 1 | 1 | 0 | 0 | 2 |
| 4 | 3 | 1 | 2 | 1 | 0 | 2 | 2 | 1 | 1 | 2 | 2 | 0 | 2 |
| ... | ... | ... | ... | ... | ... | ... | ... | ... | ... | ... | ... | ... | ... |
| 597 | 0 | 1 | 0 | 0 | 2 | 1 | 3 | 1 | 1 | 0 | 1 | 0 | 6 |
| 598 | 1 | 1 | 1 | 0 | 2 | 1 | 2 | 1 | 2 | 1 | 2 | 0 | 6 |
| 599 | 3 | 1 | 1 | 0 | 0 | 2 | 3 | 1 | 1 | 0 | 1 | 1 | 4 |
| 600 | 4 | 2 | 2 | 0 | 2 | 1 | 0 | 0 | 1 | 0 | 1 | 0 | 7 |

**Table 6.** Meaning of conditional attribute values.

| Condition Properties | Meaning of Attribute Value | | | | | |
|---|---|---|---|---|---|---|
| a1 | 0: Bottom | 1: Waterline | 2: Hull | 3: Deck | 4: Cabin | 5: Superstructure |
| a2 | 0: 0–15 (cm) | | 1: 16–25 (cm) | | 2: 26–35 (cm) | |
| a3 | 0: 0–15 (MPa) | | 1: 16–25 (MPa) | | 2: 26–35 (MPa) | |
| a4 | 0: <250 (Pa.s) | | | 1: ≥250 (Pa.s) | | |
| a5 | 0: 0–60 (μm) | | 1: 60–120 (μm) | | 2: >120 (μm) | |
| a6 | 0: Quick drying | | 1: Standard drying | | 2: Slow drying | |
| a7 | 0: Sa1 | 1: Sa2 | | 2: Sa2.5 | | 3: Sa3 |
| a8 | 0: 0~4 | | | 1: ≥5 | | |
| a9 | 0: 0–15 (°C) | | 1: 16–30 (°C) | | 2: >30 (°C) | |
| a10 | 0: <45 (%) | | 1: 45–65 (%) | | 2: >65 (%) | |
| a11 | 0: 0–10 (°C) | | 1: 11–20 (°C) | | 2: >20 (°C) | |
| a12 | 0: Group A | | 1: Group B | | 2: Group C | |

The algorithm parameter settings are shown in Table 7. In the table, $P_{c1}$ and $P_{c2}$ are crossover probability control factors, $P_{m1}$ and $P_{m2}$ are mutation probability control factors, $\alpha$ and $\lambda$ are approximate equivalence threshold and attribute number weight adjustment parameters, respectively, $n$ is population size, $m$ is the number of qubits, $\tau$ is the maximum number of iterations, $I$ is the catastrophe threshold algebra, $t_0$ is the initial temperature, $t_s$ is the termination temperature, $a$ is the cooling coefficient, $n_s$ is the number of new solutions not accepted in the simulated annealing algorithm, and $N$ is the the number of iterations per temperature.

**Table 7.** Algorithm parameters of ARIGA, QGA, and IHQGA.

| Algorithm Name | Parameter Information | | | | | | | |
|---|---|---|---|---|---|---|---|---|
| ARIGA | Parameter name | $P_{c1}$ | $P_{c2}$ | $P_{m1}$ | $P_{m2}$ | $\alpha$ | $\lambda$ | |
| | Parameter value | 0.9 | 0.6 | 0.1 | 0.001 | 0.7 | 0.35 | |
| QGA | Parameter name | $n$ | | $m$ | | $\tau$ | | $I$ |
| | Parameter value | 500 | | 12 | | 100 | | 50 |
| IHQGA | Parameter name | $n$ | $m$ | $\tau$ | $t_0$ | $t_s$ | $a$ | $n_s$ | $N$ |
| | Parameter value | 500 | 12 | 100 | 1500 | 100 | 0.99 | 50 | 200 |

(1)    Comparison of attribute reduction algorithms

① Population diversity analysis: Population diversity is a prerequisite for the evolution of genetic algorithms, and a diverse population means that the algorithm has a more robust performance. In order to analyze the population diversity of the three algorithms,

set the termination condition of the algorithm only to meet the maximum number of iterations of 100, and record the changes of population diversity of the three algorithms in the reduction process, as shown in Figure 5.

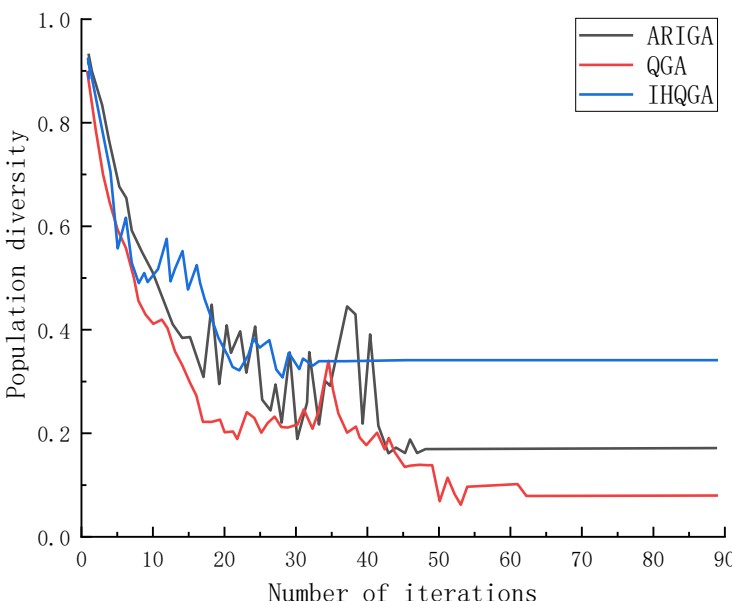

**Figure 5.** Population diversity change process.

The IHQGA algorithm uses dynamic probability amplitude values to different initial populations. It uses a simulated annealing algorithm to maintain the population diversity at a certain level to avoid premature maturity and ensure the algorithm's performance. The literature proposes that the ARIGA algorithm uses adaptive crossover and variation operators, while the population diversity during evolution fluctuates and changes unstably before it finally maintains at a low level. The traditional QGA algorithm has better population diversity than the genetic algorithm due to its unique encoding and updating method. However, it is not further improved, making the population diversity decline faster than the other two algorithms and maintaining the lowest population diversity level.

② Analysis of the superiority-seeking ability: The evolutionary process of attribute approximation of the three algorithms was selected for comparison, and the changes in the optimal fitness and average fitness values of the iterative populations of the three algorithms were recorded, as shown in Figure 6a,b.

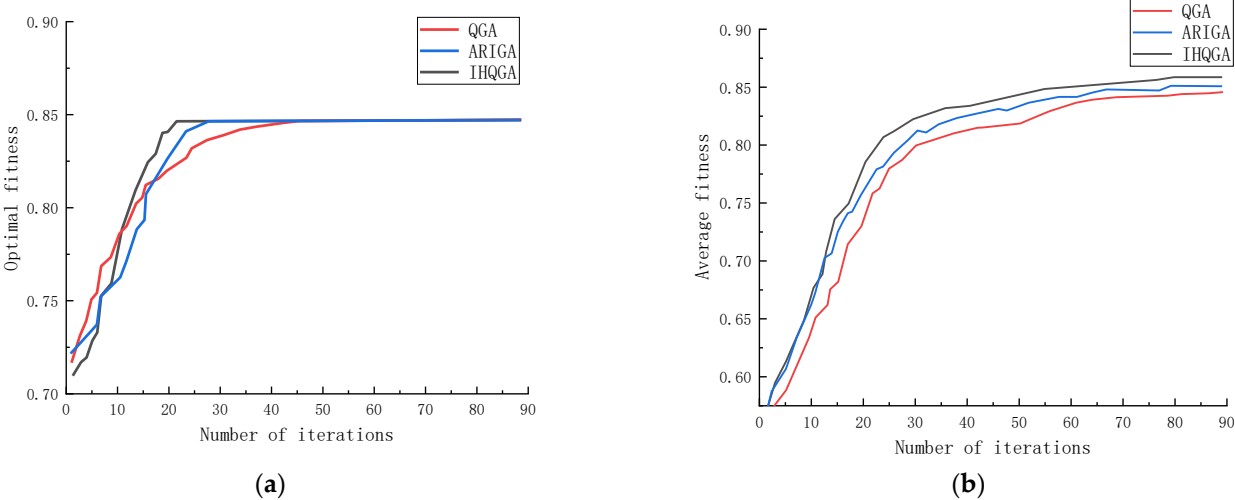

**(a)**        **(b)**

**Figure 6.** Optimal fitness and average fitness change process: (**a**) Optimal fitness; (**b**) Average fitness.

As shown from Figure 6a, QGA, ARIGA, and IHQGA reach the optimal fitness of 0.847 and begin to converge after 43, 25, and 22 iterations, respectively. IHQGA can reach optimal fitness faster. It can be seen from Figure 6b that the average fitness value of the attribute reduction algorithm based on IHQGA is higher than the other two algorithms, and the fitness change rate of the IHQGA algorithm is faster, indicating that IHQGA can quickly guide the direction of population evolution. Therefore, compared with QGA and aria, IHQGA proposed in this paper has more muscular optimization efficiency.

③ Analysis of the reduction capability: The number of attributes after reduction and the time used by the algorithm for reduction are used as comparison items, and five reduction experiments are conducted to avoid chance. The comparison results are shown in Figure 7a. Compared with QGA and ARIGA, IHQGA adopts simulated annealing for search supplementation, improving algorithm accuracy, and fewer attributes obtained are obtained. The use of adaptive magnitude and adaptive rotation angle improves the algorithm iteration speed, and the algorithm's average consumption time is 57,450 s, which proves that the algorithm has a significant effect on search efficiency and accuracy. The error analysis was performed by calculating the standard deviation of the data of these five trials, and the results are shown in Figure 7b. The error bars of ARIGA's running time are too long, and the experimental results have noticeable variability and poor reproducibility; the error bars of QGA and IHQGA are both of uniform length, with more minor errors, more stable data, less dispersion, and high confidence.

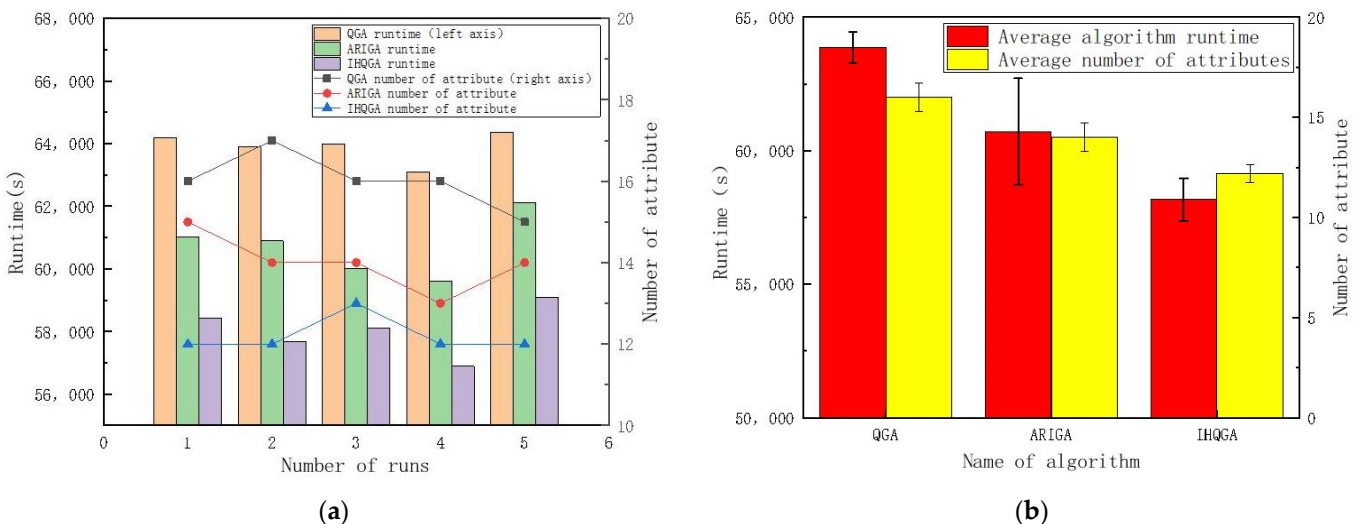

(**a**)　　　　　　　　　　　　　　　　　　　　　　　　(**b**)

**Figure 7.** Comparison of attribute reduction time and performance: (**a**) Reduction ability comparison; (**b**) Error analysis.

(2)　Analysis of knowledge acquisition results

The improved hybrid quantum genetic algorithm proposed in this paper is used for attribute reduction of decision table.,Input decision Table 5 which the number of populations is 600 and the maximum number of iterations is 100. The output of the reduced decision table is shown in Table 8. Table 8 is combined with rule extraction to obtain the generative rule set, as shown in Table 9.

**Table 8.** Minimum decision table after reduction.

| U | a1 | a2 | a4 | a5 | a6 | a10 | a11 | a12 | d |
|---|----|----|----|----|----|-----|-----|-----|---|
| 1 | 5 | 1 | 1 | 0 | 1 | 1 | 2 | 2 | 1 |
| 2 | 4 | 1 | 1 | 0 | 0 | 2 | 0 | 0 | 6 |
| 3 | 0 | 0 | 1 | 2 | 0 | 2 | 0 | 1 | 8 |
| 4 | 2 | 2 | 1 | 0 | 1 | 2 | 1 | 0 | 1 |
| 5 | 4 | 2 | 0 | 2 | 0 | 2 | 1 | 1 | 2 |
| ... | ... | ... | ... | ... | ... | ... | ... | ... | ... |
| 118 | 4 | 2 | 1 | 2 | 0 | 0 | 1 | 1 | 5 |
| 119 | 3 | 1 | 2 | 0 | 0 | 0 | 1 | 1 | 9 |
| 120 | 4 | 2 | 2 | 0 | 0 | 2 | 0 | 0 | 4 |
| 121 | 5 | 0 | 1 | 1 | 2 | 0 | 1 | 1 | 14 |
| 122 | 4 | 2 | 1 | 1 | 0 | 1 | 2 | 1 | 3 |

**Table 9.** Rules set for ship coating defects.

| Number of Rules | Rule Form |
|-----------------|-----------|
| 1 | IF a4 = 0 AND a5 = 0 AND a12 = 2 THEN d = 1 |
| 2 | IF a1 = 5 THEN d = 1 |
| 3 | IF a4 = 0 AND a5 = 1 AND a11 = 0 AND a12 = 2 THEN d = 2 |
| 4 | IF a5 = 2 AND a11 = 2 AND a12 = 1 THEN d = 2 |
| 5 | IF a2 = 0 AND THEN d = 2 |
| ... | ... |
| 31 | IF a4 = 1 AND a5 = 2 AND a10 = 2 THEN d = 17 |

The number of rules, the rule correctness rate, and the expert approval degree is selected as the criteria for judging the effectiveness of rules. The number of rules represents the amount of redundant information or the amount of omitted information. The rule correctness rate refers to the proportion of correct rules to the number of rules verified by experiments. The expert approval degree is the score of each rule by a certain number of experts to judge its applicability. The results are compared using the knowledge acquisition method based on a traditional rough set [38], the knowledge acquisition method based on the QGA optimized rough set in the literature [36], the knowledge acquisition method based on the ARIGA optimized rough set in the literature [37], and the knowledge acquisition method proposed in this paper. The work undertaken in this paper and the methods used in the literature are shown in Table 10. To ensure that the comparison results are reasonable, the above methods are tested using the same discrete method and dataset, and the knowledge acquisition results are shown in Table 11. It can be seen that the knowledge acquisition method of IHQGA-RS proposed in this paper is better than ARIGA-RS in terms of the number of rules and the rule correctness rate. Compared with RS and QGA-RS methods, the number of rules and rule correctness indexes is better. RS and QGA-RS have too much redundant information and low acquisition efficiency due to the shortcomings of their algorithms, but the recognition of acquisition results is higher. The IHQGA-RS method proposed in this paper obtains concise results, has a higher rule correctness rate, and has the highest expert approval.

*4.3. Application Example*

To demonstrate the application value of the knowledge acquisition method proposed in this paper, we applied it to the 81200DWT bulk carrier project of a shipyard in China in 2019. We compared the knowledge acquisition results with expert judgment results to test the validity of knowledge. First, we obtained data from Section 4.2 of the block coating manual (deck) numbered CX0813A-5F39003CB. There were seven defect samples in Section 4.2. The results of knowledge acquisition using the method in this paper and the final comparison with the expert diagnosis results are shown in Table 12. Through comparison, it can be found that although the method proposed in this paper has not

identified a few reasons, the results are consistent with the experts' conclusions, indicating that the knowledge discovered is beneficial to the subsequent design work of coating designers and has application value.

**Table 10.** Comparison of the work content of several knowledge acquisition methods.

| Job Description \ Methods of Knowledge Acquisition | RS | QGA-RS | ARIGA-RS | IHQGA-RS |
|---|---|---|---|---|
| Attribute reduction algorithm | Reduction algorithm based on the importance of attributes | Reduction algorithm based on traditional QGA | Reduction algorithm based on ARIGA | Reduction algorithm based on IHQGA |
| Test data | Iris | EIA | NOAA | 201KBC |
| Discrete method | Equidistance division | NaiveScaler algorithm | SemiNaiveScaler algorithm | Fuzzy C-mean clustering algorithm |
| Fields of affiliation | Life Sciences | Electrical Sciences | Geographical Sciences | Physical Sciences |
| Specific application | Species identification | Troubleshooting | Prediction | Prediction |

**Table 11.** Comparison of knowledge acquisition results.

| Index \ Methods of Knowledge Acquisition | RS | QGA-RS | ARIGA-RS | IHQGA-RS |
|---|---|---|---|---|
| Number of rules | 54 | 42 | 28 | 31 |
| Rule accuracy (%) | 81.5 | 85.0 | 92.9 | 88.2 |
| Expert recognition (score) | 80 | 90 | 50 | 95 |

**Table 12.** The defect results of this method were compared with the expert diagnosis results.

| Defect Case | Results of the Method in This Paper | The Result of Expert Diagnosis |
|---|---|---|
| 1 | If the coating part is a pipe and the wet film thickness is less than 60 (μm), there will be missed coating. | The coating part is uneven and the spraying is uneven. |
| 2 | If the derusting grade is Sa1 and the surface temperature is 0–10 (°C), fish eye defects appear. | The surface does not reach the rust removal grade. |
| 3 | | When the substrate temperature is too low, the wettability of the coating is reduced. |
| 4 | If the diluent used is quick drying type, there will be color seepage defects. | The roughness grade is not enough, the diluent volatilizes too fast, and the operation is improper. |
| 5 | If the spray gun distance is greater than 25 (cm), color seepage defects appear. | The roughness grade is not enough, the diluent volatilizes too fast, and the operation is improper. |
| 6 | If the spray gun distance is less than 15 (cm), sagging defects appear. | The spray gun is too close. |
| 7 | | The viscosity of the coating is too low. |

## 5. Conclusions

This paper introduces a knowledge acquisition of ship coating defects based on a rough set optimized by IHQGA and validates it with shipyard coating defect example data. Based on traditional QGA, a rough set-oriented IHQGA attribute reduction algorithm was proposed. On the basis of optimizing QGA probability amplitude, rotation angle parameter setting, and revolving door updating strategy, a simulated annealing algorithm was introduced to enhance its local search capability, and it was used in the rough set attribute reduction process. The performance of the improved hybrid quantum genetic algorithm proposed in this paper is tested based on the actual shipyard data, and the results show that the improved hybrid quantum genetic algorithm proposed in this paper has good search performance and optimization accuracy. It is known by comparison with the other three knowledge acquisition methods that the knowledge acquisition method proposed

in this paper has fewer attributes reduction results and higher identification accuracy than the standard rough set, the rough set knowledge acquisition method using the QGA attribute reduction algorithm, and the ARIGA attribute reduction algorithm proposed in the literature. Thus, this method is suitable for the ship coating process. Through the verification of segmented examples, it is proved that this method has high theoretical significance and practical application value.

The knowledge acquisition method of the IHQGA optimized rough set proposed in this paper adopts the basic rough set model, which itself encounters the disadvantages of noise, missing values, large data volume, continuous attributes that need to be discretized, etc., causing the actual effect to still not reach the expectation, and the later stage will consider introducing the extended model of the rough set model or combining other theories such as fuzzy mathematics and neural network. For the IHQGA attribute simplification algorithm proposed in this paper, the performance of the algorithm has been greatly improved, but there is still room for improvement. Later, we will consider changing its coding method or adopting a new quantum gate update method to further improve the performance of the algorithm.

**Author Contributions:** H.B. revised the paper and completed it; X.J. wrote the first draft of the paper; J.Z. and H.L. assisted in the experimental verification of the paper; X.Y. and B.P. collected and sorted the data; H.Z. provided financial assistance, provided experimental sites, and thesis methods. All authors have read and agreed to the published version of the manuscript.

**Funding:** This study was financially supported by Ministry of Industry and Information Technology High-Tech Ship Research Project: Research on Development and Application of Digital Process Design System for Ship Coating (No.: MC-202003-Z01-02), the National Natural Science Foundation of China (No.:51804133) and the Natural Science Foundation of Jiangsu Province (No.: BK20180977).

**Institutional Review Board Statement:** Not applicable.

**Informed Consent Statement:** Not applicable.

**Data Availability Statement:** The data presented in this study are available on request from the corresponding author. The data are not publicly available because they are also part of ongoing research.

**Conflicts of Interest:** The authors declare that there is no conflict of interest regarding the publication of this work.

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
