# Peer review of "A Knowledge Acquisition Method of Ship Coating Defects Based on IHQGA-RS"

_coatings, doi:10.3390/coatings12030292_

Round 1

Reviewer 1 Report

Authors have improved the paper significantly, therefore paper can be accepted for publication.

I suggest authors to make the figure captions more descriptive.

Reviewer 2 Report

This paper proposes  a knowledge acquisition method based on a rough set optimized by an improved hybrid quantum genetic algorithm to guide a construction process of ship coating. A set of previously collected ship coating data has been used to validate the proposed method. Results shows an improvement in the efficiency of the process. My previous minor concerns on this paper have been addressed by the authors. I would suggest authors to proofread the paper.

Reviewer 3 Report

Authors have proposed a knowledge acquisition method based on a rough set (RS) optimized by an improved hybrid quantum genetic algorithm (IHQGA) to guide the ship coating construction process. This is an interesting article and the algorithm has been tested good size of data with the reduction of consumption time.  Moreover, Authors have claimed that the knowledge acquisition result based on the IHQGA optimization rough set has 20%~50% fewer rules and 5%~10% higher accuracy than other methods. Following comments could be considered before accepting this article for final publication.

  1. Authors have discussed in section 2 about knowledge acquisition method based on Rough Set Theory, It is suggested that authors should give proper references for each statement. 
  2. Authors should give proper evidence based statement supported by references when they explain about different methods and also IHQGA. 
  3.  Heading of subsection 4.1 is written as  Date preparation, it should be Data Preparation. 
  4.  Authors have claimed that the source of the experimental data set is the painting examples of 210 KBC bulk carrier painting process using NJ6528 high-pressure airless spraying machine in a shipyard between 2015 and 2020. However, proper reference should be provided for this. 
  5. Authors have used acronyms in tables, they should provide the full names in caption or in tables.
  6. Authors should make a table to compare the results of their proposed method and the work reported in literature or even done by themselves with existing algorithms. 

Reviewer 4 Report

This is a model work of application of marine coatings on ship body. I see very minimal application value in the presented data and do not recommend its publication. The manuscript also needs a through language edit. I suggest the authors relate more their mathematical data to actual painting scenario and marine coatings.

Author Response

This is a model work of application of marine coatings on ship body. I see very minimal application value in the presented data and do not recommend its publication. The manuscript also needs a through language edit. I suggest the authors relate more their mathematical data to actual painting scenario and marine coatings.

Point 1: This is a model work of application of marine coatings on ship body. I see very minimal application value in the presented data and do not recommend its publication.

Response 1: Thank you very much for your review. This article mainly obtains knowledge from the historical data in the shipyard. According to our research in domestic shipyards, there is no special prevention work for coating defects, which leads to the fact that the staff only repair when there are defects. If we can prevent it in coating design and construction, we can reduce time and cost. Therefore, this work still has certain use value. We can quantify the causes of coating defects according to previous cases to help the coating work of subsequent ship projects. Please give us another chance to provide suggestions for our paper. Thank you very much.

Point 2: The manuscript also needs a through language edit.

Response 2: According to your suggestion, we have re edited the English language of the article. If we still need to make language changes, we will use the polishing service provided by the journal.

Point 3: I suggest the authors relate more their mathematical data to actual painting scenario and marine coatings.

Response 3: According to your suggestion, we have added a section about the example verification of a ship section. It is a small-scale experiment we did last year. Our knowledge acquisition results are basically consistent with those obtained by experts in the shipyard, which shows that our method is practical.

The revised contents are as follows:

4.3. Application example

To demonstrate the application value of the knowledge acquisition method proposed in this paper, we applied it to the 81200DWT bulk carrier project of a Shipyard in China in 2019. We compared the knowledge acquisition results with expert judgment results to test the validity of knowledge. First, we obtained data from section 502 of block coating manual (deck) numbered CX0813A-5F39003CB. There were 7 defect samples in section 502. The results of knowledge acquisition using the method in this paper and the final comparison with the expert diagnosis results are shown in Table 12. Through comparison, it can be found that although the method proposed in this paper has not identified a few reasons, the results are consistent with the experts' conclusions, indicating that the knowledge discovered is beneficial to the subsequent design work of coating designers and has application value.

Table 12. The defect results of this method were compared with the expert diagnosis results.

Defect case

Results of the method in this paper

The result of expert diagnosis

1

If the coating part is a pipe and the wet film thickness is less than 60 (µ m), there will be missed coating.

The coating part is uneven and the spraying is uneven.

2

If the derusting grade is Sa1 and the surface temperature is 0 ~ 10 (℃), fish eye defects appear.

The surface does not reach the rust removal grade.

3

When the substrate temperature is too low, the wettability of the coating is reduced.

4

If the diluent used is quick drying type, there will be color seepage defects.

The roughness grade is not enough, the diluent volatilizes too fast, and the operation is improper.

5

If the spray gun distance is greater than 25 (cm), color seepage defects appear.

The roughness grade is not enough, the diluent volatilizes too fast, and the operation is improper.

6

If the spray gun distance is less than 15 (cm), sagging defects appear.

The spray gun is too close.

7

The viscosity of the coating is too low.

Special thanks to you for your good comments. We have tried our best to improve the manuscript and made some changes in the manuscript. These changes will not influence the content and framework of the paper, and the changes we did not list here have been marked up using the “Track Changes” in revised paper. We appreciate Editors/Reviewers' warm work earnestly, and hope the correction will meet with approval. Once again, thank you very much for your comments and suggestions.

Reviewer 5 Report

The present paper is interesting without any doubt. Overall, the manuscript is well organized. It contains interesting results that are worth to be published. Accordingly, the scientific message is conveyed with clarity which makes this publication of great value. Thus, I recommend the present paper for publication in Coatings, just very minor modifications should be performed:

The paper contains some minor grammatical errors and typo-mistakes that should be corrected.

Also, the reported results have been well described and interpreted. However, I would recommend providing some experimental (practical) results if possible.

What about the effect of "corrosion"? because the system could be in different corrosive environments. How the environmental changes were taken into consideration.

Comparisons with traditional ways should be performed.

Round 2

Reviewer 4 Report

Yes, the paper is now acceptable.

Regards

This manuscript is a resubmission of an earlier submission. The following is a list of the peer review reports and author responses from that submission.

Round 1

Reviewer 1 Report

Figure 7. Comparison of attribute reduction time and performance. Author should conduct error analysis and add error bars.

The references are cited in the text without adding space. Please add space before and after reference number.

Figure 3. principle of QGA combined with SA. The presentation of figure is not good, it is difficult to under stand the scheme. Please make it more descriptive.

Figure 1. Ship painting operation site. I hope these images are owned by the authors otherwise take permission of reusing these figures.

Some typo and grammatical errors are noticed which should be removed during revision.

In conclusions, authors should add future aspects of this work.

It is suggested to make abstract more quantitative.

Reviewer 2 Report

This paper seems incomplete, because it doesn't present CONCLUSIONS.

Moreover, my recommendation id REJECT based on the following reasons:

  1. The scientific interest of the work is reduced, in my opinion;
  2. The quality of presentation is low, lacking full stops, spaces before the references, and so on;
  3. There are sentences without sense. Just one example in the Introduction: "The coating of anti-corrosion coatings is ...";
  4. The number of references is low than expected and about 1/3 of them are more than 10 years old;
  5. The section Materials & Methods is very ill-described and the starting point seems to be closer to Literature Review than Methods. Moreover, subsection 2.1.1 is not Methods, should be traferred to the Literature Review;:
  6. Moreover, 2.1.2. seems a set of different topics without a proper flow. Moreover, some topics seems not directly related to the names used in Figure 2;
  7. The IHQGA methodology should be properly introduced, which is not the case;
  8. The flowchart of Figure 4 is not devided into steps, as the explanation, which cause a lack of explanations and flow of information;
  9. Sometimes, the units are ill-represented, as in Table 1, where "MPa" appears described as "Mpa" (the same error apears in Table 4). Moreover, in the same table, it is impossible to understand the last line. Other variables, as "Evaporation rate", the classification is qualitative (merged with others in quantitative form), which is not correct;;
  10. The units are bonded to the values, against the standard rules;
  11. The formulation of the algorithm is not clear;
  12. The discussion doesn't use any reference. Thus, this is not a discussion, but a summary of the work. However, the Discussion is missing, comparing with other simiar works.

Reviewer 3 Report

This paper proposes  a knowledge acquisition method based on a rough set optimized by an improved hybrid quantum genetic algorithm to guide a construction process of ship coating. A set of previously collected ship coating data has been used to validate the proposed method. Results shows an improvement in the efficiency of the process. Generally, the paper is well written. Please revise the paper based on following comments

A lot of developments of robotic and automation solutions for ship coating maintenance could be seen in literature. Authors are suggested to discuss these developments in the literature review.

E.g.,

Fernández-Isla, Carlos, Pedro J. Navarro, and Pedro María Alcover. "Automated visual inspection of ship hull surfaces using the wavelet transform." Mathematical Problems in Engineering 2013 (2013).

Muthugala, MA Viraj J., SM Bhagya P. Samarakoon, and Mohan Rajesh Elara. "Toward energy-efficient online Complete Coverage Path Planning of a ship hull maintenance robot based on Glasius Bio-inspired Neural Network." Expert Systems with Applications 187 (2022): 115940.

Li, Xue, et al. "A semi-automatic system for grit-blasting operation in shipyard." 2018 IEEE 23rd International Conference on Emerging Technologies and Factory Automation (ETFA). Vol. 1. IEEE, 2018.

Mention the statistical tests used for the evaluating the significance of comparison results. Also provide the test statistics such as P values.

Proof read the paper to correct typos and grammatical error. e.g., Correct Fig 1 caption.

Round 2

Reviewer 2 Report

No comments.